# Snow avalanche deaths in Switzerland from 1995 to 2014—Results of a nation-wide linkage study

Claudia Berlin[1], Frank Techel[2,3], Beat Kaspar Moor[4], Marcel Zwahlen[1], Rebecca Maria Hasler[1,5,6]*, for the Swiss National Cohort study group[¶]

**1** Institute of Social and Preventive Medicine, University of Bern, Bern, Switzerland, **2** WSL Institute for Snow and Avalanche Research SLF, Davos, Switzerland, **3** University of Zürich, Department of Geography, Zürich, Switzerland, **4** Department of Orthopaedic Surgery and Traumatology, Hôpital du Valais, Martigny, Switzerland, **5** Department of Emergency Medicine, Inselspital, University Hospital Bern, Bern, Switzerland, **6** Department of Trauma, University Hospital Zurich, Zurich, Switzerland

¶ Membership of the Swiss National Cohort study group is provided in the Acknowledgments.
* rebecca.hasler@insel.ch

**Data Availability Statement:** Individual data from different datasets were used for the construction of the SNC. All these data are the property of the

## Abstract

### Objectives

More than 20 people die each year in snow avalanches in Switzerland. Previous studies have primarily described these victims, but were not population based. We investigated sociodemographic factors for avalanche mortality between 1995 and 2014 in the entire Swiss resident population.

### Design and methods

Within the Swiss National Cohort we ascertained avalanche deaths by anonymous data linkage with the avalanche accident database at the Swiss WSL Institute of Snow and Avalanche Research SLF. We calculated incidence rates, by dividing the number of deaths from avalanches by the number of person-years, and hazard ratios (HRs) for sociodemographic and economic characteristics using Cox proportional hazard models.

### Results

The data linkage yielded 250 deaths from avalanche within the SNC population for the 20 years 1995 to 2014. The median distance between the place of residence and the place of the event (avalanche) was 61.1 km. Male gender, younger age (15–45 years), Swiss nationality, living in the Alpine regions, higher education, living in the highest socioeconomic quintile of neighbourhoods, being single, and living in a household with one or more children were associated with higher avalanche mortality rates. Furthermore, for younger persons (<40 years) the hazard of dying in an avalanche between 2005 and 2014 was significantly lower than in the years 1995 to 2004 (HR = 0.56, 95%-CI: 0.36–0.85).

Swiss Federal Statistical Office (SFSO) and can only be made available by legal agreements with the SFSO. This also applies to derivatives such as the analysis files used for this study. Therefore, the SNC and researchers working with SNC data are not allowed to make SNC data sets publicly available, especially on person level. However, after approval of the SNC Scientific Board, a specific SNC module contract with SFSO would allow researchers to receive analysis files for replication of the analysis. Data requests should be sent to Prof. Milo Puhan (chairman of the SNC Scientific Board, miloalan.puhan@uzh.ch). The data extracted from the SLF database are available upon request (data@slf.ch), with reference to the title of the study.

**Funding:** RMH was supported by the Swiss National Science Foundation (grant nos. 3347CO-108806, 33CS30_134273 and 33CS30_148415) to conduct this small nested project. The SNC Scientific Board approved the implementation of this study but had no role in study design, data collection and analysis, decision to publish, or preparation of the manuscript.

**Competing interests:** The authors have declared that no competing interests exist.

**Abbreviations:** 95% CI, 95% Confidence Interval; CR, Crude Rate; HR, Hazard Ratio; ICD-10, International Classification of Disease, Injuries and Causes of Death, 10th revision; SEP, Socioeconomic position; SLF, WSL Institute for Snow and Avalanche Research SLF; SNC, Swiss National Cohort; SSEP, Swiss neighbourhood index of socioeconomic position.

## Conclusion

Over a 20-year period in Switzerland, higher rates of dying in an avalanche were observed in men, in younger age groups, and persons with tertiary education, living in the highest socioeconomic quintile of neighbourhoods, and living in an Alpine region. For younger persons (<40 years), the hazard declined during the study period.

## Introduction

Snow avalanches claim the lives of about 100 people each year in the European Alps [1]. About one-quarter of these deaths occur in Switzerland, where at least 200 persons are caught annually by more than 100 avalanches [2]. More than 90% of the victims lose their lives during recreational activities away from avalanche-secured areas [2]. A large proportion of deaths occur in the cantons of Valais and Grisons, which are known for ski touring and can have an unfavourable snowpack structure that might lead to more accidental avalanches [3]. In recent decades, substantial efforts have been devoted to preventing avalanche deaths, by increasing avalanche awareness and by use of specialised rescue equipment, including avalanche transceivers and airbag-systems [4–7].

Avalanche victims are typically characterized in accident statistics by age and gender, sometimes by nationality, and rarely by region of residence [8–10]. However, more detailed sociodemographic information about avalanche victims may help direct prevention programs to specific risk groups. Most studies of accident events do not look in detail at the at-risk or source groups. Approaches to consider these include extracting usage statistics from heli-ski logs, national park registrations, social media platforms, representative population surveys, or simply by counting people [11–14].

The Swiss National Cohort (SNC) is a cohort study of the entire Swiss-resident population that has allowed investigating cause-specific mortality rates [15,16]. We linked SNC death information with reported avalanche accidents to investigate whether the mortality rates for dying in an avalanche differ by sociodemographic factors and the relationship between place of residence and location of accident.

## Data and methods

### WSL Institute for Snow and Avalanche Research SLF (SLF)

The SLF documents deaths caused by snow avalanches, excluding deaths due to falling ice and snow sliding off buildings. Avalanches are reported by a dense network of observers and by rescue services, and may also be reported by cantonal authorities and members of the general public. Additionally, for accidents resulting in deaths, detailed police investigation reports are generally available [17].

Data for all victims of avalanches between 1 January 1995 and 31 December 2014 were retrieved from SLF's database including information about cause of death, date of the avalanche accident, gender, year of birth, canton/country of residence, and nationality. The activity of the accident party at the time of the avalanche event also was extracted. Within this last group, we differentiated backcountry touring activities on skis or snowshoes from off-piste riding in unsecured terrain close to ski areas, based on the often detailed police investigation and rescue reports.

Due to incomplete data on nationality, canton/country of residence, year of birth, and gender, it was not always possible to distinguish between victims who were Swiss, foreigners living in Switzerland, and foreigners from abroad.

## Swiss National Cohort

The SNC is a longitudinal study of mortality in Switzerland containing sociodemographic and economic information for the entire population of Switzerland. Owing to the lack of a unique person identifier, census data from 1990 and 2000 were linked to death or migration records using deterministic and probabilistic linkage methods based on sex, date of birth, place of residence, nationality, marital status, religion, and profession. From 2010 onwards, the census is performed yearly, data are registry-based, and information from different registries can be linked via a unique person identifier. The SNC database follows mortality and migration up to 2014. More information about the SNC is given elsewhere [15,16,18].

## Ethical approval and consent to participate

Approval for the Swiss National Cohort study and a data center established at ISPM Bern was obtained from the Ethics Committees of the Cantons of Zurich and Bern. For this type of study, formal consent is not required. SNC and SLF data were fully anonymized.

## SLF and SNC data linkage

The SLF database does not include socio-demographic data such as household type, marital status, neighbourhood index or education. Furthermore, the ICD 10 code X36 does not identify precisely just snow avalanche deaths in the SNC data. SNC data does not include type of activity when the avalanche happened. Therefore, we linked SLF and SNC data on Swiss and foreign inhabitants in Switzerland, identifying avalanche victims based on sex, year of birth, date of death/avalanche, nationality, canton of residence, and cause of death. The Swiss death certificate allows the recording of initial disease, a consecutive disease, and two concomitant diseases, which are used to determine the primary cause of death. The diseases and cause of death are coded using the International Classification of Disease, Injuries and Causes of Death, 10th revision (ICD-10).[19] ICD-10 has been used in Swiss death certificates since 1 January 1995. We restricted the study to the calendar years beginning and after 1995 in which the same ICD-10 coding was in use. The ICD-10 code X36 records a "victim of avalanche, landslip or other movement of soil". We searched for X36 codes in all available disease or cause of death variables to identify avalanche victims, and used the variables mentioned above to link SLF avalanche victims to SNC avalanche victims. The linkage results indicated expansion of the search to include additional codes (W02, W15, W17, W77, X31, X59, and Y86; see S1 Table for explanation; see also WHO ICD-10 classification: http://apps.who.int/classifications/icd10/browse/2010/en#), and broadening the range between the day of the avalanche and the date of death (adding the days buried under snow, or 40 days) since not every avalanche victim dies immediately, or the day of finding the dead person has been recorded as the date of death. We also allowed for variation in the year of birth by ±1 year.

The characteristics of our study population were obtained at two times, in the 1990 and the 2000 census (see Fig 1 in [15]). Except for age, we assumed characteristics did not change from 1990 to 1995. Between 1990 and 2000, people could have been born, died, or migrated into or out of Switzerland. Consequently, not all persons appear again in the census 2000 and new persons have been included. We used the latest sociodemographic information available (either 1990 or 2000) and mortality information through the end of 2014. In Table 1 we present the

**Table 1. Characteristics of the study population at 1 January 1995 and 5 December 2000 and the 250 avalanche deaths and rates per 1 million population occurring during the periods 1 January 1995 to 4 December 2000, and from 5 December 2000 to 31 December 2014.**

| Characteristics | Dec 1995 | | Dec 1995 –Nov 2000 | | Dec 2000 | | Dec 2000 –Dec 2014 | | Avalanche victims (overall) | |
|---|---|---|---|---|---|---|---|---|---|---|
| | Population | | Avalanche vicitims | | Population | | Avalanche vicitims | | Crude rate per 1000000 | |
| | Freq. | Percent | Freq. | Percent | Freq. | Percent | Freq. | Percent | Rate | 95% CI |
| Total | 6516102 | 100.0 | 72 | 100.0 | 7280041 | 100.0 | 178 | 100.0 | 1.93 | 1.70–2.18 |
| **Sex** | | | p<0.0000 | | | | p<0.0000 | | | |
| Male | 3201750 | 49.1 | 65 | 90.3 | 3563896 | 49.0 | 152 | 85.4 | 3.41 | 2.99–3.90 |
| Female | 3314352 | 50.9 | 7 | 9.7 | 3716145 | 51.0 | 26 | 14.6 | 0.50 | 0.35–0.70 |
| **Age** | | | p<0.0000 | | | | p<0.0000 | | | |
| <15 | 805966 | 12.4 | 4 | 5.6 | 1207802 | 16.6 | 19 | 10.7 | 1.67 | 1.25–2.23 |
| 15–24 | 795624 | 12.2 | 25 | 34.7 | 848718 | 11.7 | 36 | 20.2 | 4.12 | 3.23–5.26 |
| 25–34 | 1121669 | 17.2 | 14 | 19.4 | 1059796 | 14.6 | 48 | 27.0 | 2.61 | 2.03–3.34 |
| 35–44 | 1047261 | 16.1 | 10 | 13.9 | 1201443 | 16.5 | 40 | 22.5 | 1.84 | 1.35–2.52 |
| 45–54 | 961149 | 14.8 | 12 | 16.7 | 1005390 | 13.8 | 23 | 12.9 | 1.41 | 0.95–2.08 |
| 55–64 | 718497 | 11.0 | 6 | 8.3 | 808560 | 11.1 | 9 | 5.1 | 0.98 | 0.56–1.73 |
| 65+ | 1065936 | 16.4 | 1 | 1.4 | 1148332 | 15.8 | 3 | 1.7 | 0.09 | 0.01–0.61 |
| **Education** | | | p<0.4352 | | | | p<0.0000 | | | |
| Compulsory education or less, not known | 3110399 | 47.7 | 29 | 40.3 | 3149550 | 43.3 | 45 | 25.3 | 1.39 | 1.11–1.75 |
| Upper secondary level education | 2739930 | 42.0 | 34 | 47.2 | 2987240 | 41.0 | 81 | 45.5 | 2.06 | 1.72–2.47 |
| Tertiary level education | 665773 | 10.2 | 9 | 12.5 | 1143251 | 15.7 | 52 | 29.2 | 2.92 | 2.27–3.75 |
| **Nationality** | | | p<0.0836 | | | | p<0.0001 | | | |
| Swiss | 5378114 | 82.5 | 65 | 90.3 | 5779574 | 79.4 | 163 | 91.6 | 2.16 | 1.90–2.46 |
| Non-Swiss | 1137988 | 17.5 | 7 | 9.7 | 1500467 | 20.6 | 15 | 8.4 | 0.91 | 0.60–1.38 |
| **Marital status** | | | p<0.0000 | | | | p<0.0000 | | | |
| Single | 2817112 | 43.2 | 50 | 69.4 | 3058321 | 42.0 | 102 | 57.3 | 2.86 | 2.44–3.35 |
| Married/Widowed/Divorced | 3698990 | 56.8 | 22 | 30.6 | 4221720 | 58.0 | 76 | 42.7 | 1.28 | 1.05–1.56 |
| **Type of household** | | | p<0.0083 | | | | p<0.0231 | | | |
| Single person household | 1214521 | 18.6 | 10 | 13.9 | 1631727 | 22.4 | 40 | 22.5 | 1.73 | 1.31–2.28 |
| Couple without children | 1434958 | 22.0 | 6 | 8.3 | 1730000 | 23.8 | 26 | 14.6 | 1.02 | 0.72–1.44 |
| Couple with 1 or more children | 3489403 | 53.6 | 49 | 68.1 | 3531076 | 48.5 | 103 | 57.9 | 2.39 | 2.04–2.81 |
| Others | 377220 | 5.8 | 7 | 9.7 | 387238 | 5.3 | 9 | 5.1 | 2.66 | 1.63–4.34 |
| **Region** | | | p<0.0000 | | | | p<0.0000 | | | |
| *Swiss alpine regions* | | | | | | | | | | |
| Eastern Alps | 157683 | 2.4 | 1 | 1.4 | 178145 | 2.4 | 16 | 9.0 | 5.36 | 3.33–8.62 |
| Southern Alps | 273991 | 4.2 | 1 | 1.4 | 314240 | 4.3 | 4 | 2.2 | 0.89 | 0.37–2.14 |
| Western Alps | 201259 | 3.1 | 10 | 13.9 | 229701 | 3.2 | 33 | 18.5 | 10.46 | 7.76–14.11 |
| Northern Alps | 1030793 | 15.8 | 15 | 20.8 | 1164732 | 16.0 | 44 | 24.7 | 2.83 | 2.19–3.65 |
| *Swiss non-alpine regions* | | | | | | | | | | |
| ≤ 25km from the Northern Alps | 2062229 | 31.6 | 14 | 19.4 | 2298583 | 31.6 | 33 | 18.5 | 1.14 | 0.86–1.52 |
| > 25km from the Northern Alps | 2790147 | 42.8 | 31 | 43.1 | 3094640 | 42.5 | 48 | 27.0 | 1.44 | 1.15–1.79 |
| **Urbanization** | | | p<0.8487 | | | | p<0.0197 | | | |
| urban | 1952189 | 30.0 | 21 | 29.2 | 2075785 | 28.5 | 50 | 28.1 | 1.94 | 1.54–2.45 |
| periurban | 2854920 | 43.8 | 30 | 41.7 | 3263827 | 44.8 | 65 | 36.5 | 1.63 | 1.33–1.99 |
| rural | 1708993 | 26.2 | 21 | 29.2 | 1940429 | 26.7 | 63 | 35.4 | 2.41 | 1.95–2.99 |
| **Religious affiliation** | | | p<0.7202 | | | | p<0.0027 | | | |
| Protestant | 2655775 | 40.8 | 32 | 44.4 | 2567354 | 35.3 | 61 | 34.3 | 1.99 | 1.62–2.44 |
| Roman Catholic | 3002792 | 46.1 | 33 | 45.8 | 3045762 | 41.8 | 85 | 47.8 | 2.14 | 1.79–2.56 |

*(Continued)*

**Table 1.** (Continued)

| Characteristics | Dec 1995 | | Dec 1995 –Nov 2000 | | Dec 2000 | | Dec 2000 –Dec 2014 | | Avalanche victims (overall) | |
| --- | --- | --- | --- | --- | --- | --- | --- | --- | --- | --- |
| | Population | | Avalanche vicitims | | Population | | Avalanche vicitims | | Crude rate per 1000000 | |
| | Freq. | Percent | Freq. | Percent | Freq. | Percent | Freq. | Percent | Rate | 95% CI |
| No religious affiliation | 487246 | 7.5 | 5 | 6.9 | 809255 | 11.1 | 26 | 14.6 | 2.19 | 1.54–3.12 |
| Other/ unknown | 370289 | 5.7 | 2 | 2.8 | 857670 | 11.8 | 6 | 3.4 | 0.58 | 0.29–1.16 |
| **Swiss neighbourhood index of SEP** | | | p<0.0300 | | | | p<0.3644 | | | |
| Lowest quintile | 1618864 | 24.8 | 20 | 27.8 | 1688519 | 23.2 | 45 | 25.3 | 2.17 | 1.70–2.77 |
| Second quintile | 1341649 | 20.6 | 11 | 15.3 | 1427617 | 19.6 | 30 | 16.9 | 1.60 | 1.18–2.18 |
| Third quintile | 1253474 | 19.2 | 9 | 12.5 | 1362470 | 18.7 | 25 | 14.0 | 1.39 | 0.99–1.94 |
| Fourth quintile | 1179493 | 18.1 | 11 | 15.3 | 1319232 | 18.1 | 31 | 17.4 | 1.76 | 1.30–2.38 |
| Highest quintile | 1050945 | 16.1 | 21 | 29.2 | 1190818 | 16.4 | 35 | 19.7 | 2.60 | 2.00–3.38 |
| Missing | 71677 | 1.1 | 0 | 0.0 | 291385 | 4.0 | 12 | 6.7 | - | - |
| **Language region** | | | p<0.0218 | | | | p<0.0065 | | | |
| German | 4728695 | 72.6 | 45 | 62.5 | 5241390 | 72.0 | 115 | 64.6 | 1.70 | 1.46–1.99 |
| French | 1507980 | 23.1 | 26 | 36.1 | 1718485 | 23.6 | 59 | 33.1 | 2.81 | 2.27–3.47 |
| Italian | 279427 | 4.3 | 1 | 1.4 | 320166 | 4.4 | 4 | 2.2 | 0.88 | 0.36–2.10 |

characteristics of our study population in 1995 and 2000, and report avalanche mortality rates for the whole study period 1995–2014.

We calculated crude rates (CR) by dividing the number of avalanche deaths by the number of person-years (per 1 000 000). To compare individuals with different sociodemographic and economic characteristics, we fitted Cox proportional hazard models and estimated corresponding hazard ratios (HR), which express the ratios of event hazards of compared groups. Hazard can be seen as an instantaneous event rate defined as the probability of an event occurring in the next time interval, standardized by the length of that interval [20]. For the time-to-event analyses, the time at risk per person starts on 1 January 1995 or 5 December 2000 and ends on the date of loss to follow-up at 4 December 2000, migrating out of Switzerland, death, or 31 December 2014, whichever occurred first. We used individual's age as time axis, that is, the age at which a person entered the study and the age at which the avalanche death occurred, or observation stopped. With the choice of this time axis, Cox proportional hazard regression models provide hazard ratios that are automatically age-adjusted. To account for calendar effects, we divided the time of observation into an early (1995–2004) and a later period (2005–2014). We also assessed interactions between the covariates.

We included the following sociodemographic and economic characteristics in our analyses: sex, age, education (compulsory education or less, not known; upper secondary education; tertiary level education), nationality (Swiss, non-Swiss), marital status (single, married/widowed/divorced), type of household (single-person household, couple without children, couple with one or more children, others), urbanization level of the municipality of residence (urban, peri-urban, rural), religious affiliation (Protestant, Roman Catholic, no religious affiliation, other/unknown), and language region (German, French, Italian). We introduced a region variable dividing Switzerland into four Alpine regions (Western, Southern, Eastern, and Northern Alps, as in Techel et al.[3]), and two regions defined by the distance to the Alps (an area within 25 km of the northern border of the Alps and an area more than 25 km away from the northern border of the Alps, see Fig 1). Finally, we used the Swiss neighbourhood index of

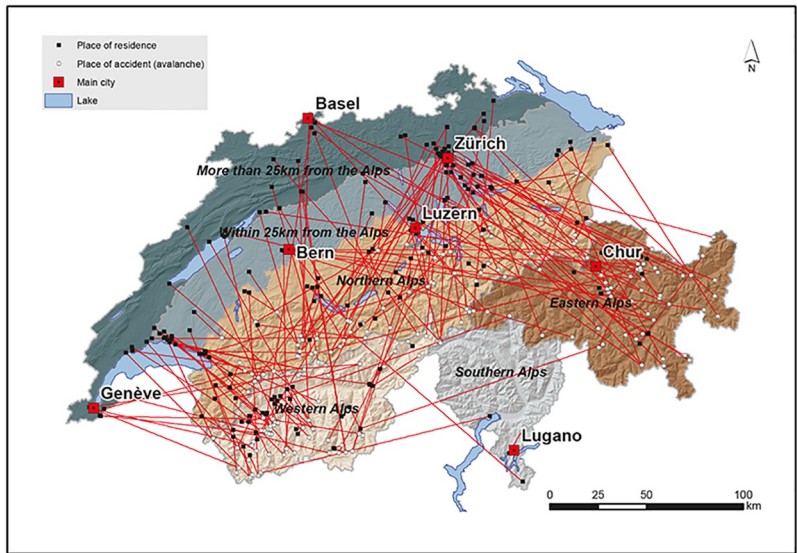

**Fig 1. Map of Switzerland illustrating the regional variable and places of residence and of avalanche accidents.**

socioeconomic position (SSEP), divided into quintiles, that has been constructed based on several variables surveyed at the censuses [21].

Statistical analyses were carried out with Stata 15 (Stata Corporation, College Station, Texas, USA). We used the STROBE cohort reporting guidelines [22].

## Results

Between 1995 and 2014, 439 died (range 12–34 annually) after being caught by an avalanche. Excluding 157 persons who were foreigners who lived abroad, 282 persons from the SLF database were possible linkage candidates. The flowchart in Fig 2 summarizes how we finally identified 250 avalanche victims by linking SLF records to the SNC database.

Table 1 lists characteristics of the study population at the two time points 1 January 1995 and 5 December 2000 and the 250 avalanche deaths for the two time periods from the 1 January 1995 to 4 December 2000, and from 5 December 2000 to 31 December 2014. We report overall mortality rates per 1 million person-years Most avalanche victims were overwhelmingly male, in their twenties and thirties, resided in the non-Alpine region more than 25 km from the Northern Alps, and lived in the German language region. The crude death rate for males was almost 7 times higher than that for females. Swiss citizens had a crude rate more than double that of non-Swiss residents. The geographical region with the highest death rate of 10.46 per 1 million person-years (95% CI 7.76–14.11) was the Western Alps. The crude rate for residents of the Swiss Alpine regions was 4.23 per 1 million person-years (95% CI 3.54–5.06) about 3 times higher than for residents of non-Alpine regions.

We then restricted our Cox regression analyses to 6 639 174 persons 15 years or older with information on education and available SSEP information (see Fig 2). In this restricted population, there were 215 avalanche victims. Among these 215 victims, 143 died while backcountry touring, 59 died off-piste riding, and 13 died in a building, skiing on an open ski run, or while travelling on a public transportation corridor. A sensitivity analysis excluding these 13 persons gave almost identical results.

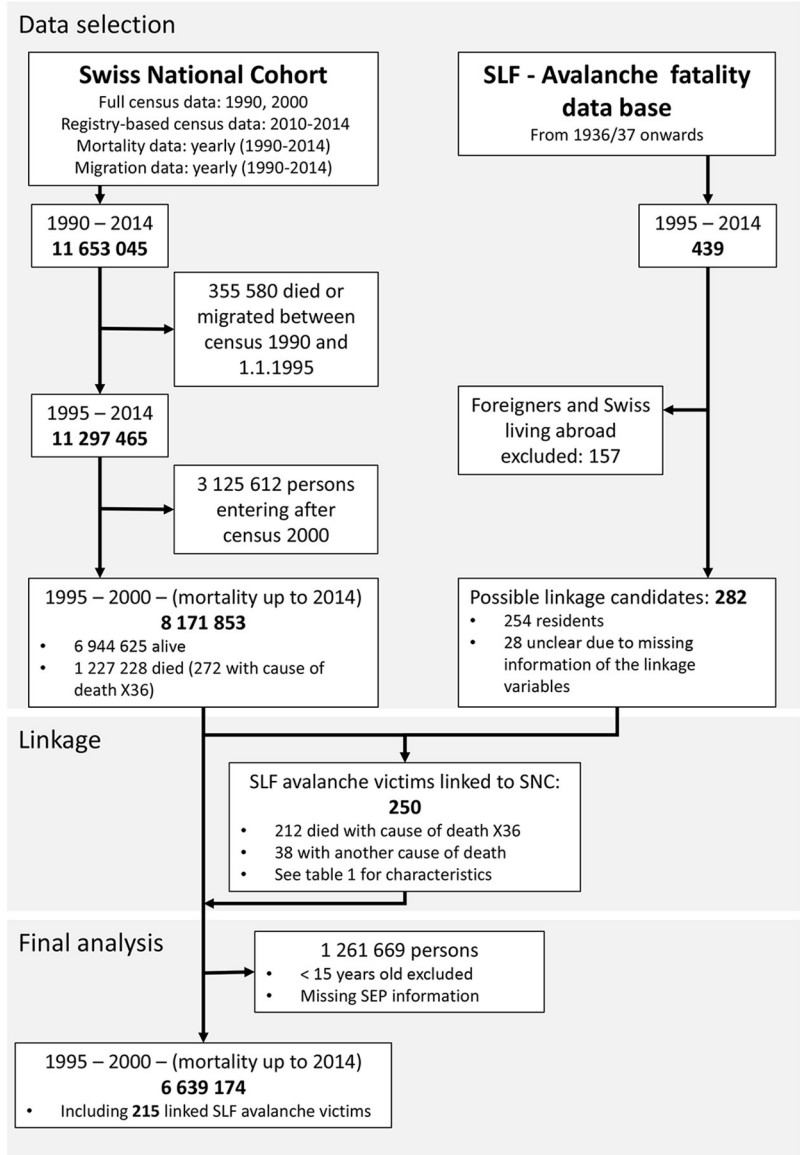

**Fig 2. Flowchart illustrating the linkage process and selection of the study participants for the final analysis.**

The median linear distance between the place of residence and the place of the avalanche accident was 61.1 km, ranging from 1.2 km to 285.3 km (interquartile range 20–111.2 km). Not surprisingly, this distance was significantly larger for those living outside the Alps or in the Southern Alps (median 79–118 km) compared to those who lived in the other Alpine regions (p-value < .00001). Residents in the Eastern, Western, and Northern Alps generally died close to their place of residence (median 12–31 km), and within their Alpine region (84%, see S2 Table). Fig 1 illustrates places of residence and of avalanche accidents.

We omitted language region in the Cox regression analyses since it is highly collinear with region of residence (the Italian language region and the Southern Alps are almost identical). Because we found significant interactions between age and education, we conducted separate analyses for the younger (15–39 years) and older (40+ years) age groups.

Table 2 displays the results of the Cox analyses of the overall, younger, and older cohorts. The pronounced reduced risk for women of dying in an avalanche compared to men persisted in all three adjusted Cox models (overall HR = 0.18, 95% CI 0.12–0.26). Non-Swiss had a lower hazard than Swiss residents did in the overall analysis (HR = 0.54, 95%-CI 0.32–0.90). The hazard associated with living in one of the three Alpine regions was clearly higher in the overall analyses than living in the region more than 25 km away from the northern border of the Alps. These associations were similarly strong in the younger and older age groups.

Persons 40 years or older with an upper secondary or tertiary level of education had a higher hazard of avalanche death than those with only compulsory education or less. The hazard did not seem to differ by the level of urbanization of the municipality of residence. However, in the overall analysis the hazard for persons living in neighbourhoods with an SSEP in the highest quintile was higher that of persons in neighbourhoods in the lowest quintile; the result was more pronounced in the older age group. When comparing the years 2005–2014 to 1995–2004, we found a reduced hazard in the younger cohort, but not the older, during the later period.

## Discussion

Over a period of 20 years (1995–2014), the rate of dying in an avalanche was clearly lower in the 2nd decade in those younger than 40 years. We observed increased rates for males, and for persons with an education beyond the compulsory minimum level, being single, and living in a household with one or more children. With a median distance between place of death and place of residence of 61.1 km, most victims died not far from home which was reflected in a higher rate for those living in Alpine areas.

### Strengths and limitations

This is the first study to analyse the risk to different groups in the Swiss population of dying in an avalanche. The ICD-10 code X36 defines not only snow avalanches but also death from landslides or other earth movements. A major strength of our study derives from our linkage of SNC and SLF data that teased out snow avalanche deaths, alone, in the SNC data. This direct linkage provided detailed socio-demographic data, such as household type, marital status, socioeconomic data or education and allowed us to connect avalanche victims' places of residence with the locations of their fatal accidents.

Among the limiting factors in this study is that some of the sociodemographic indicators that were derived at the time of each census, e.g., marital status, type of household and SSEP, may have changed. Also, due to incomplete information in the SLF database not all avalanche deaths recorded could be linked to the SNC statistics. Finally, Swiss residents may also die in avalanches outside Switzerland; this appears to be true for about 10% of all Swiss avalanche victims [11].

To strengthen prevention efforts, a better knowledge of the activity profiles of participants in these outdoor activities would be needed [8–14]. The SLF accident statistics and the SNC data are clearly limited in this regard.

### Interpretation

The profile of avalanche victims in this study might primarily echo the profile of ski tourers and off-piste skiers in general. We noted that victims residing in the Alps died close to their place of residence (12–31 km distance). We could therefore speculate that this might be a typical traveling distance between place of residence and place of recreation, but also that many of these victims were likely rather familiar with that region and its specific circumstances.

**Table 2. Results of the Cox regression analyses for dying in an avalanche for the overall, younger, and older cohorts.**

| Characteristics | All persons | | 15–39 yrs | | 40+ yrs | |
|---|---|---|---|---|---|---|
| | Hazard ratio | 95% CI | Hazard ratio | 95% CI | Hazard ratio | 95% CI |
| **Sex** | | **p<0.001** | | **p<0.001** | | **p<0.001** |
| Male | 1 | | 1 | | 1 | |
| Female | 0.178 | 0.120–0.263 | 0.172 | 0.0997–0.297 | 0.201 | 0.115–0.352 |
| **Education** | | **p = 0.274** | | **p = 0.138** | | **p = 0.012** |
| Compulsory education or less, not known | 1 | | 1 | | 1 | |
| Upper secondary level education | 1.176 | 0.805–1.718 | 0.711 | 0.452–1.120 | 2.943 | 1.139–7.608 |
| Tertiary level education | 1.432 | 0.915–2.240 | 0.512 | 0.258–1.019 | 4.146 | 1.560–11.02 |
| **Nationality** | | **p = 0.018** | | **p = 0.051** | | **p = 0.243** |
| Swiss | 1 | | 1 | | 1 | |
| Non-Swiss | 0.536 | 0.319–0.900 | 0.483 | 0.233–1.004 | 0.648 | 0.312–1.344 |
| **Marital status** | | **p<0.001** | | **p<0.001** | | **p = 0.350** |
| Single | 1 | | 1 | | 1 | |
| Married/Widowed/Divorced | 0.419 | 0.279–0.627 | 0.182 | 0.0918–0.359 | 0.732 | 0.381–1.407 |
| **Type of household** | | **p = 0.014** | | **p = 0.482** | | **p = 0.010** |
| Single person household | 1 | | 1 | | 1 | |
| Couple without children | 0.937 | 0.581–1.511 | 1.506 | 0.817–2.777 | 0.637 | 0.301–1.347 |
| Couple with 1 or more children | 1.645 | 1.119–2.419 | 1.393 | 0.856–2.265 | 1.698 | 0.896–3.220 |
| Others | 1.210 | 0.607–2.409 | 1.118 | 0.486–2.574 | 1.546 | 0.450–5.312 |
| **Region** | | **p<0.001** | | **p<0.001** | | **p<0.001** |
| *Swiss alpine regions* | | | | | | |
| Eastern Alps | 3.855 | 2.099–7.079 | 2.452 | 1.026–5.862 | 6.358 | 2.691–15.02 |
| Southern Alps | 0.631 | 0.195–2.044 | 0.620 | 0.146–2.632 | 0.638 | 0.084–4.815 |
| Western Alps | 9.913 | 6.120–16.06 | 7.291 | 3.856–13.79 | 14.90 | 7.111–31.24 |
| Northern Alps | 2.399 | 1.643–3.503 | 1.545 | 0.915–2.607 | 3.944 | 2.253–6.906 |
| *Swiss non-alpine regions* | | | | | | |
| ≤ 25km from the Northern Alps | 0.852 | 0.579–1.253 | 0.618 | 0.364–1.052 | 1.237 | 0.697–2.195 |
| > 25km from the Northern Alps | 1 | | 1 | | 1 | |
| **Urbanization** | | **p = 0.739** | | **p = 0.839** | | **p = 0.583** |
| urban | 1 | | 1 | | 1 | |
| periurban | 0.928 | 0.661–1.303 | 1.049 | 0.655–1.682 | 0.795 | 0.489–1.294 |
| rural | 1.065 | 0.724–1.566 | 1.168 | 0.688–1.981 | 0.985 | 0.562–1.726 |
| **Religious affiliation** | | **p = 0.187** | | **p = 0.163** | | **p = 0.691** |
| Protestant | 1 | | 1 | | 1 | |
| Roman Catholic | 0.777 | 0.566–1.068 | 0.755 | 0.490–1.163 | 0.823 | 0.515–1.316 |
| No religious affiliation | 0.834 | 0.520–1.338 | 0.601 | 0.290–1.244 | 1.108 | 0.590–2.084 |
| Other/unknown | 0.477 | 0.222–1.029 | 0.352 | 0.120–1.038 | 0.669 | 0.227–1.973 |
| **Swiss neighbourhood index of SEP** | | **p<0.001** | | **p = 0.325** | | **p = 0.001** |
| Lowest quintile | 1 | | 1 | | 1 | |
| Second quintile | 0.791 | 0.514–1.217 | 0.733 | 0.415–1.295 | 0.831 | 0.428–1.614 |
| Third quintile | 0.879 | 0.555–1.393 | 0.841 | 0.461–1.531 | 0.869 | 0.424–1.781 |
| Fourth quintile | 1.300 | 0.827–2.043 | 1.125 | 0.617–2.051 | 1.427 | 0.717–2.841 |
| Highest quintile | 2.072 | 1.312–3.273 | 1.419 | 0.754–2.671 | 2.784 | 1.407–5.509 |
| **Observation interval** | | **p = 0.930** | | **p = 0.007** | | **p = 0.238** |
| 1995–2004 | 1 | | 1 | | 1 | |
| 2005–2014 | 0.987 | 0.738–1.320 | 0.556 | 0.361–0.854 | 1.281 | 0.849–1.933 |

Successful prevention of deaths in avalanches in Switzerland needs to be tailored to patterns of winter sport activities of the Swiss resident population. Information on these activities exists and comes from different sources, and studies conducted by the Federal Office for Sports [23–25] and the Swiss Council for Accident Prevention [26]. In recent years, the number of persons pursuing winter sports in unsecured terrain has increased [11]. Results from surveys in the year 2014 suggest that approximately 2% of the population do backcountry tours [23] and that about a quarter are riding off-piste at times when skiing in terrains accessible from ski areas [23,26]. Persons riding off-piste are more often male [23,26] and 15 to 29 years old [26], while persons undertaking ski tours and snowshoeing are generally older [23]. We observed similar patterns for avalanche mortality rates (see S3 and S4 Tables).

The median age at death of the avalanche victims in our analysis increased from 31 years in the first ten years of our study period (1995–2004: N = 123) to 42 years in the last ten years of our study period (2005–2014: N = 127). This is also reflected in the hazard for younger persons (15–39 years), which was significantly lower for the years 2005–2014 than for the years 1995–2004. Whether this is linked to an increased use of avalanche safety gear in this age group, changes in risk behaviour, or whether fewer younger people participate in these outdoor-activities, is unclear. However, similar ageing trends have also been noted for the users of the Swiss avalanche forecast, but also for avalanche victims in France [9,27].

As only a small proportion of the resident population is exposed to avalanche hazard, we suggest that contrasting the socio-economic profiles of avalanche victims with that of the population in general, is one way to explore patterns in avalanche victims' profiles. However, as has been the case in many previous studies (e.g. [9]) the lack of knowledge concerning the true population at risk of dying in an avalanche also impacts the interpretation of our findings.

## Conclusion

Over a 20 year period in Switzerland, higher rates of dying in an avalanche were observed in men, in younger age groups, and persons with tertiary education, living in the highest socio-economic quintile of neighbourhoods, and living in an Alpine region. However, for younger persons (<40 years), the rate declined during the study period.

## Supporting information

**S1 Table. Additional used ICD 10 codes indicating the cause of death.**
(PDF)

**S2 Table. Place of residence and place of avalanche of the 215 avalanche victims by region (1995–2014).**
(PDF)

**S3 Table. Results of the Cox regression analyses for dying in an avalanche while undertaking backcountry touring on skis or snowshoes for the overall, younger, and older cohorts.**
(PDF)

**S4 Table. Results of the Cox regression analyses for dying in an avalanche while riding off-piste for the overall, younger, and older cohorts.**
(PDF)

**S1 Checklist. STROBE cohort checklist.**
(DOCX)

## Acknowledgments

We thank the Swiss Federal Statistical Office for providing mortality and census data and for the support (in developing and implementing the idea of the SNC), which made the Swiss National Cohort and this study possible. The members of the Swiss National Cohort Study Group are Matthias Egger (Chairman of the Executive Board), Adrian Spoerri and Marcel Zwahlen (all Bern), Milo Puhan (Chairman of the Scientific Board), Matthias Bopp (both Zurich), Martin Röösli (Basel), Michel Oris (Geneva) and Murielle Bochud (Lausanne). We thank Christopher Ritter for his editorial assistance.

## Author Contributions

**Conceptualization:** Marcel Zwahlen, Rebecca Maria Hasler.

**Data curation:** Claudia Berlin, Frank Techel.

**Formal analysis:** Claudia Berlin.

**Funding acquisition:** Rebecca Maria Hasler.

**Investigation:** Claudia Berlin.

**Methodology:** Marcel Zwahlen, Rebecca Maria Hasler.

**Project administration:** Marcel Zwahlen.

**Resources:** Marcel Zwahlen.

**Supervision:** Marcel Zwahlen, Rebecca Maria Hasler.

**Validation:** Marcel Zwahlen.

**Visualization:** Claudia Berlin.

**Writing – original draft:** Claudia Berlin, Frank Techel, Marcel Zwahlen, Rebecca Maria Hasler.

**Writing – review & editing:** Claudia Berlin, Frank Techel, Beat Kaspar Moor, Marcel Zwahlen, Rebecca Maria Hasler.

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
