## [Decision Letter · Decision Letter 0]

21 Aug 2019

PONE-D-19-17260

Snow avalanche deaths in Switzerland from 1995 to 2014 – results of a nation-wide linkage study

PLOS ONE

Dear Dr. Berlin,

Thank you for submitting your manuscript to PLOS ONE. After careful consideration, we feel that it has merit but does not fully meet PLOS ONE’s publication criteria as it currently stands. Therefore, we invite you to submit a revised version of the manuscript that addresses the points raised during the review process.

Two reviewers with expertise in the field have evaluated your manuscript and provided very useful comments to build a stronger paper. Among these comments, please pay particular attention to (1) clarifying the segment presenting the statistical analysis, and how the  cohorts were built, (2) highlighting differences between individuals involved in skiing and those dying in avalanches, if possible, and how the linkage with the entire population might contribute to developing preventive measures, (3) removing or refining the distinction between voluntary and involuntary exposure.

We would appreciate receiving your revised manuscript by Oct 05 2019 11:59PM. To enhance the reproducibility of your results, we recommend that if applicable you deposit your laboratory protocols in protocols.io, where a protocol can be assigned its own identifier (DOI) such that it can be cited independently in the future. For instructions see: http://journals.plos.org/plosone/s/submission-guidelines#loc-laboratory-protocols

We look forward to receiving your revised manuscript.

Kind regards,

Bruno Masquelier, PhD

Academic Editor

PLOS ONE

Journal Requirements:

2. In the ethics statement in the manuscript and in the online submission form, please provide additional information about the patient records used in your retrospective study.

Specifically, please ensure that you have discussed whether all data were fully anonymized before you accessed them and/or whether the IRB or ethics committee waived the requirement for informed consent.

If patients provided informed written consent to have data from their medical records used in research, please include this information.

"This work was supported by the Swiss National Science Foundation (grant nos. 3347CO-108806, 33CS30_134273 and 33CS30_148415)."

" RMH received 30.000 CHF from the SNC to conduct this small nested project. The SNC Scientific Board had no role in study design, data collection and analysis, decision to publish, or preparation of the manuscript."

Please provide an amended Funding Statement that declares *all* the funding or sources of support received during this specific study (whether external or internal to your organization) as detailed online in our guide for authors at http://journals.plos.org/plosone/s/submit-now  Please ensure you state the name(s) of your funder(s) in full. Please state what role the funders took in the study.  If any authors received a salary from any of your funders, please state which authors and which funder. If the funders had no role, please state: "The funders had no role in study design, data collection and analysis, decision to publish, or preparation of the manuscript."

5. Please include captions for your Supporting Information files at the end of your manuscript, and update any in-text citations to match accordingly. Please see our Supporting Information guidelines for more information: http://journals.plos.org/plosone/s/supporting-information

Reviewers' comments:

Reviewer's Responses to Questions

**Comments to the Author**

1. Is the manuscript technically sound, and do the data support the conclusions?

Reviewer #1: Partly

Reviewer #2: Yes

2. Has the statistical analysis been performed appropriately and rigorously? 

Reviewer #1: I Don't Know

Reviewer #2: I Don't Know

3. Have the authors made all data underlying the findings in their manuscript fully available?

Reviewer #1: Yes

Reviewer #2: Yes

4. Is the manuscript presented in an intelligible fashion and written in standard English?

Reviewer #1: Yes

Reviewer #2: Yes

5. Review Comments to the Author

Reviewer #1: This paper tackles an important topic that has been poorly investigated until now, mostly because of methods constraints and difficulties.

In order to help direct prevention towards specific risk groups, the authors intend to provide detailed sociodemographic information about avalanche victims in Switzerland: do some sociodemographic factors impact exposure to avalanche mortality?

The originality of the approach lies in the linking of 2 databases: the "Snow and Avalanche Research" (SLF) which lists deaths caused by avalanches in Switzerland; the Swiss National Cohort (SNC) which reports the broader causes of mortality in Switzerland.

The authors highlight three main results.

The most interesting one is the connection revealed between avalanche victims’ places of residence and the locations of their fatal accidents. This is an important insight that should be developed further, in terms of interpretation, for the paper to provide a significant contribution.

The second result is the rate of 2 fatalities in avalanche for 1 million swiss residents each year. This information does not fit the research's objective and does not provide helpful information in terms of prevention, since it does not make it possible to estimate the risk exposure of actual mountain sports practitioners.

The third result is a very general profile of avalanche victims: mostly (young) men, with tertiary education, a privileged economic situation, living in an Alpine region. This result is not key since this merely echoes the profile of ski tourers and off-piste skiers. In other words, the methods used led to confirm that peoppe dying in avalanches are people involved in those dangerous activities, which is tautologic.

As a consequence, linking specific data (SLF) to global ones (SNC) does not prove relevant regarding the objective stated in the introduction: basing prevention on a more detailed picture of victims socio-demographics.

In order to get their paper published, the authors should address a couple of concerns, and acknowledge some limitations of their study:

- the methodological and scientific justification of the linkage between the databases must be explained more thoroughly. What does it make possible? What does it add to SLF data, in concrete terms? Currently, it is not celar why this linkage is useful and what limitation it makes it possibe to overcome.

- in the introduction, the authors state that « none of these studies allows a direct linkage between those recreating and those becoming a victim of an avalanche ». The problem is, broadening the scope to the swiss population does not help neither. By doing so, the authors don't focus on those recreating, which is the gap they identified in the literature. It must be made clearer why it helps to turn to general population data.

According to prevention experts, what is needed to enhance prevention is less a comparison with general population, than a better knowledge of the plural profiles of participants in the dangerous activities.

Beyond this central remark, some minor issues (easier to address) can be pointed out:

- the methods used by SLF to gather information regarding avalanches and avalanche victims must be precised. It will make it easier to understand if the data are precise and exhaustive. A better knowledge of such criteria would also help to understand the facts that are included or excluded in the data base

- what does « to be caught in an avalanche » mean? There are many envisageable outputs while an avalanche occurs: being swept away in the avalanche, and/or being buried in snow, and/or getting injured in tha avalanche, and/or being rescued (by specialized rescuers or members of the group), etc. This point is all the most crucial to address since it is further stated that 10% of the population that gets « caught » dies because of the avalanche. More precision is needed here.

- The authors write that « the activity of the accident party at the time of the avalanche event also was extracted » : is it always possible to identify the activity in question ? Especially when differentiating between off-piste skiing and ski touring: can we be positive on this distinction? What is it based on?

- « Persons voluntarily expose themselves to avalanche risk » : it seems more appropriate to replace voluntarily by knowingly

- Regarding the ICD-10 code X36: sometimes, while reporting the cause of deaths, physicians only use generic physiological terms. How can we be sure that this seemingly precise codification is properly used?

- « We speculate that persons with higher education, which often goes along with higher socioeconomic status and higher income, have more resources for leisure activities like ski-touring » : it seems simplistic to link a priviledged position, on an economical plan, and involvement in ski touring. The diffusion of this technical, distinctive activity is also, not to say mostly, a social and cultural process

- Some data from Switzerland or France highlight that over time, people dying in avalanche are older and older (from 37 to 42 for example, in France, in 25 or 30 years). This is a point that should be mentioned in the paper

- Last, avalanche safety equipment should be quantified; the reader actually needs a figure of equipment rate, and type of equipment. By the way, it seems that avalanche airbags are more commonly used in Switzerland than in Italy or France

Reviewer #2: This paper presents the most comprehensive review of Swiss avalanche fatality statistics to date and provides details of sociodemographic status and place of residence not previously reported. Additionally, larger trends of reduced fatalities in younger age groups over time is demonstrated raising some hope that avalanche safety technology and education may be successful.

The statistical analysis is complex and difficult to follow as currently written in the methods and results and clarification is needed. That being said, the discussion and conclusions follow logically from the stated objectives and the overall paper adds to the knowledge base regarding avalanche fatalities in Switzerland.

Line:

47- “still, 10%...die” needs reference.

72- Data regarding cause of avalanche death, time of burial, not relevant to study objective. Not reported in results or discussion. Eliminate.

76- One could argue that any skiing in mountainous terrain, “avalanche-secured” or otherwise, is a voluntary exposure to avalanches. Skiing is clearly a different exposure than occupying buildings or moving through transportation corridors. Your project does not address risk taking behavior specifically and voluntary/involuntary distinction as you describe does not change results. Would remove or re-word the voluntary/involuntary definitions.

91- add comma after nationality

96- SLF and SNC data linkage-

The statistical analysis and linkage is complex but the description of how and why two cohorts were developed (as shown in figure 2) does not follow from the methods. Was the 1995 census data extrapolated directly from the 1990 census? Why were two cohorts, 1990-2014 and 1995-2014, chosen? It seems that there is a significant overlap between the two cohorts. One would think that the Swiss population between 1990-2014, understanding influences from deaths and migration, is a single cohort. Overall the methods section needs to be clarified and the explanation simplified in regards to the time periods evaluated and then carried over with consistent time periods to the results.

108- would like to see a few examples of the expanded ICD9 codes.

142- “two regions defined by distance from the Alps…” Using the <=>25 km distinction adds little to the demographic data as you found a median distance of 61 km from residence to place of death and described victims as predominantly from an Alpine region. This would suggest that all of Switzerland is an Alpine region, which may be true. The 25 km measure offers little useful detail beyond just place of residence and location of death.

159- Table 1. Two time periods 1/1/1995-2000 and 5/12/1995-2014 are given but it is not clear why or how these two periods were chosen. See above for line 96, there is a lack of clarity on how cohorts and time periods were determined. You also list “time periods” in line 160 then report in line 172-3 “two points in time”, both for same table. This is confusing.

6. PLOS authors have the option to publish the peer review history of their article (what does this mean?). If published, this will include your full peer review and any attached files.

Reviewer #1: No

Reviewer #2: No

---

## [Author Response · Author response to Decision Letter 0]

4 Oct 2019

Revision letter

We would like to thank both reviewers for their thoughtful comments to our manuscript. 

Please find a point by point discussion below.

Reviewer 1:

1) This paper tackles an important topic that has been poorly investigated until now, mostly because of methods constraints and difficulties. 

In order to help direct prevention towards specific risk groups, the authors intend to provide detailed sociodemographic information about avalanche victims in Switzerland: do some sociodemographic factors impact exposure to avalanche mortality?

The originality of the approach lies in the linking of 2 databases: the "Snow and Avalanche Research" (SLF) which lists deaths caused by avalanches in Switzerland; the Swiss National Cohort (SNC) which reports the broader causes of mortality in Switzerland.

Authors` response: Thank you! 

2) The authors highlight three main results.

The most interesting one is the connection revealed between avalanche victims’ places of residence and the locations of their fatal accidents. This is an important insight that should be developed further, in terms of interpretation, for the paper to provide a significant contribution.

Authors` response: We have added the following statement to the Discussion section (page 14):

« The profile of avalanche victims in this study might primarily echo the profile of ski tourers and off-piste skiers in general. We noted that victims residing in the Alps died close to their place of residence (12 – 31 km distance). We could therefore speculate that this might be a typical traveling distance between place of residence and place of recreation, but also that many of these victims were likely rather familiar with that region and its specific circumstances.«

The second result is the rate of 2 fatalities in avalanche for 1 million swiss residents each year. This information does not fit the research's objective and does not provide helpful information in terms of prevention, since it does not make it possible to estimate the risk exposure of actual mountain sports practitioners.

Authors` response: This information has been deleted in the Abstract, as well as in the manuscript (Results section, page 9 and Discussion section, page 13).

The third result is a very general profile of avalanche victims: mostly (young) men, with tertiary education, a privileged economic situation, living in an Alpine region. This result is not key since this merely echoes the profile of ski tourers and off-piste skiers. In other words, the methods used led to confirm that peoppe dying in avalanches are people involved in those dangerous activities, which is tautologic.

As a consequence, linking specific data (SLF) to global ones (SNC) does not prove relevant regarding the objective stated in the introduction: basing prevention on a more detailed picture of victims socio-demographics.

Authors` response: The authors agree that the profile of avalanche victims in this study might echo the profile of ski tourers and off-piste skiers in general. However, this has not been investigated by a population-based study before. Furthermore, previous studies do not provide detailed socio-demographic data, such as household type, marital status, neighbourhood index or education.

The following statement has been added to the Discussion section (page 13):

«This direct linkage provided detailed socio-demographic data, such as household type, marital status, socioeconomic data or education and allowed us to connect avalanche victims' places of residence with the locations of their fatal accidents.«

 And on page 14:

«The profile of avalanche victims in this study might echo the profile of ski tourers and off-piste skiers in general.«

3) In order to get their paper published, the authors should address a couple of concerns, and acknowledge some limitations of their study:

- The methodological and scientific justification of the linkage between the databases must be explained more thoroughly. What does it make possible? What does it add to SLF data, in concrete terms? Currently, it is not celar why this linkage is useful and what limitation it makes it possibe to overcome.

Authors` response: The SLF database just record avalanche deaths and accidents but does not include socio-demographic data such as household type, marital status, neighbourhood index or education. Furthermore, the ICD 10 code X36 does not identify precisely just snow avalanche deaths in the SNC data. SNC data does not include type of activity when the avalanche happened. Therefore, the data linkage has been performed.

The following statement has been added to the Methods section (page 6):

« The SLF database does not include socio-demographic data such as household type, marital status, neighbourhood index or education. Furthermore, the ICD 10 code X36 does not identify precisely just snow avalanche deaths in the SNC data. SNC data does not include type of activity when the avalanche happened. Therefore, we linked SLF recorded deaths and SNC data […]«

- in the introduction, the authors state that « none of these studies allows a direct linkage between those recreating and those becoming a victim of an avalanche ». The problem is, broadening the scope to the swiss population does not help neither. By doing so, the authors don't focus on those recreating, which is the gap they identified in the literature. It must be made clearer why it helps to turn to general population data.

Authors` response: This sentence has been deleted from the Introduction section (page 3). Please, also see our statement to your previous comment.

4) According to prevention experts, what is needed to enhance prevention is less a comparison with general population, than a better knowledge of the plural profiles of participants in the dangerous activities.

Authors` response: The following statement has been added to the Discussion section (page 13):

« To strengthen prevention efforts, a better knowledge of the activity profiles of participants in these outdoor activities would be needed (8–10,19–22). The SLF accident statistics and the SNC data are clearly limited in this regard.«

5) Beyond this central remark, some minor issues (easier to address) can be pointed out:

- the methods used by SLF to gather information regarding avalanches and avalanche victims must be precised. It will make it easier to understand if the data are precise and exhaustive. A better knowledge of such criteria would also help to understand the facts that are included or excluded in the data base

Authors` response: We explain the reporting system of the SLF at the beginning of data and methods (starting in line 77). SLF’s avalanche accident database is cross-checked annually for missing accidents or details with the mountain accident statistic by the Swiss Alpine Club. The latter receives the incidence reports of all the accidents, when the alpine rescue services were on site and therefore has a rather complete database. The number of variables describing avalanche accidents in SLF’s database is large, though the information is often incomplete. However, accidents resulting in fatalities are generally well documented. Variables, which were relevant for this study, have been listed. We added the following statement to the data description on page 5.

“Additionally, for accidents resulting in deaths, detailed police investigation reports are generally available (13).”

6) what does « to be caught in an avalanche » mean? There are many envisageable outputs while an avalanche occurs: being swept away in the avalanche, and/or being buried in snow, and/or getting injured in tha avalanche, and/or being rescued (by specialized rescuers or members of the group), etc. This point is all the most crucial to address since it is further stated that 10% of the population that gets « caught » dies because of the avalanche. More precision is needed here.

Authors’ response: As “caught in an avalanche” counts when a person is swept away by an avalanche (the person can’t ski or snowboard in a controlled way out of the avalanche). As a result, the person is either partially or fully buried, or remains on the snow surface. As “caught” counts also when a vehicle or train is hit by an avalanche and there are people inside. - It is of note, that an unknown number of less severe avalanche incidents go unreported. Therefore the 10% is an upper limit and reflects the known proportion of the people who were caught in avalanches. The most recent numbers, always for 20 years, are shown in a table published in the annual report by SLF (e.g. Zweifel et al., 2016). We have removed the 10% from the abstract and introduction, and rephrased accordingly.

7) The authors write that « the activity of the accident party at the time of the avalanche event also was extracted » : is it always possible to identify the activity in question ? Especially when differentiating between off-piste skiing and ski touring: can we be positive on this distinction? What is it based on?

Authors` response: The activity of the accident party is based on information of the SLF database. The key criteria for this distinction are whether the accident party accessed the backcountry from ski areas ascending by means of ski lifts or cable cars with only short additional hikes, or whether they ascended primarily by hiking up. For fatal accidents SLF has rather detailed police investigation reports available, generally allowing this distinction. We added the following statement in the respective section (page 5):

“Within this last group, we differentiated backcountry touring activities on skis or snowshoes from off-piste riding in unsecured terrain close to ski areas, based on the often detailed police investigation and rescue reports.” 

8) « Persons voluntarily expose themselves to avalanche risk » : it seems more appropriate to replace voluntarily by knowingly

Authors` response: Thank you for this comment. This part has been completely removed from the Methods section (page 5). See also comment 5, Reviewer 2.

9) Regarding the ICD-10 code X36: sometimes, while reporting the cause of deaths, physicians only use generic physiological terms. How can we be sure that this seemingly precise codification is properly used?

Authors` response: You are right. The X36 code can also be used for non-avalanche victims as we pointed out on page 5. The linkage of the SNC and SLF databases has also been performed to identify just snow avalanche victims. Only victims, who could be linked / identified as avalanche victims, were included in the analysis. 

10) «We speculate that persons with higher education, which often goes along with higher socioeconomic status and higher income, have more resources for leisure activities like ski-touring » : it seems simplistic to link a priviledged position, on an economical plan, and involvement in ski touring. The diffusion of this technical, distinctive activity is also, not to say mostly, a social and cultural process

Authors` response: We deleted this sentence (page 14). 

11) Some data from Switzerland or France highlight that over time, people dying in avalanche are older and older (from 37 to 42 for example, in France, in 25 or 30 years). This is a point that should be mentioned in the paper

Authors` response: Thank you for this comment. We checked whether this is true for the 250 avalanche victims we analyzed. The median age at death for the 123 persons died between 1995 – 2004 is 31 years and the median age for the 127 persons died between 2005 – 2014 increased to 42 years. We agree that over the last decades, avalanche victims are getting older and older. This information has been added to the Discussion section (page 15). We also reference two studies, who observed similar trends.

“The median age at death of the avalanche victims in our analysis increased from 31 years in the first ten years of our study period (1995-2004: N=123) to 42 years in the last ten years of our study period (2005-2014: N=127). This is also reflected in the hazard for younger persons (15-39 years), which was significantly lower for the years 2005-2014 than for the years 1995-2004. Whether this is linked to an increased use of avalanche safety gear in this age group, changes in risk behaviour, or whether fewer younger people participate in these outdoor-activities, is unclear. However, similar ageing trends have also been noted for the users of the Swiss avalanche forecast, but also for avalanche victims in France (9,27).”

12) Last, avalanche safety equipment should be quantified; the reader actually needs a figure of equipment rate, and type of equipment. By the way, it seems that avalanche airbags are more commonly used in Switzerland than in Italy or France.

Authors` response: Unfortunately, we were not able to find exact information about changes in equipment use. Some newspaper articles emphasize increased availability and sales for ski touring equipment, and mainly avalanche airbags since 2010. 

Reviewer 2:

1) This paper presents the most comprehensive review of Swiss avalanche fatality statistics to date and provides details of sociodemographic status and place of residence not previously reported. Additionally, larger trends of reduced fatalities in younger age groups over time is demonstrated raising some hope that avalanche safety technology and education may be successful.

Authors` response: Thank you!

2) The statistical analysis is complex and difficult to follow as currently written in the methods and results and clarification is needed. That beeing said, the discussion and conclusions follow logically from the stated objectives and the overall paper adds to the knowledge base regarding avalanche fatalities in Switzerland.

Authors` response: The statistical analysis is a standard time-to-event analysis in a cohort study. However, the situation is somewhat complicated by the fact that the SNC is not a closed cohort but based on 2 census rounds, one in 1990 and 2000 with “new” persons entering the cohort in 2000. We discuss this in more detail in our response to point 7) of reviewer 2.

3) Line: 47- “still, 10%...die” needs reference.

Authors’ response: We removed the 10% statement. It is based on the annual reports by SLF. These reports state, however, that particularly concerning less severe accidents, there is an unknown number of unreported cases.

4) 72- Data regarding cause of avalanche death, time of burial, not relevant to study objective. Not reported in results or discussion. Eliminate.

Authors` response: This information has been deleted in the Methods section (page 4). 

5) 76- One could argue that any skiing in mountainous terrain, “avalanche-secured” or otherwise, is a voluntary exposure to avalanches. Skiing is clearly a different exposure than occupying buildings or moving through transportation corridors. Your project does not address risk taking behavior specifically and voluntary/involuntary distinction as you describe does not change results. Would remove or re-word the voluntary/involuntary definitions.

Authors` response: Thank you. This part has been removed from the manuscript (Methods section, page 5).

6) 91- add comma after nationality

Authors` response: Thank you. This has been added. 

7) 96- SLF and SNC data linkage-

The statistical analysis and linkage is complex but the description of how and why two cohorts were developed (as shown in figure 2) does not follow from the methods. Was the 1995 census data extrapolated directly from the 1990 census? Why were two cohorts, 1990-2014 and 1995-2014, chosen? It seems that there is a significant overlap between the two cohorts. One would think that the Swiss population between 1990-2014, understanding influences from deaths and migration, is a single cohort. Overall the methods section needs to be clarified and the explanation simplified in regards to the time periods evaluated and then carried over with consistent time periods to the results.

Authors` response: Thank you for this comment. The SNC is a cohort study, which, at each of the censuses, includes new persons not seen in the previous census. The starting census was in 1990 with 6.8 Mio persons and the census in 2000 was based on 7.3 Mio persons (see cohort profile 2009). The SNC population was then followed for mortality until 2014 (including cause of death information, ICD 10 codes). Therefore we splitted the follow-up time into two time periods before and after 2000 to accommodate the new persons from the 2000 census. 

The reason to start follow-up in 1995 was the fact that ICD 10 codes have been introduced in Switzerland in 1995 to code diseases or accidents on the death certificates. Before 1995, ICD 8 codes have been used and it is even more difficult to identify snow avalanche victims using ICD 8 codes. Consequently, we report in table 1 in column “1995” the socio-demographic data of the census 1990 and the people still alive in 1995. 

We do report the data of both censuses to show how the cohort/population characteristics have changed and how the cohort grew over time. 

8) 108- would like to see a few examples of the expanded ICD9 codes.

Authors` response: We report the additional ICD 10 codes (besides ICD 10 codes X36: victim of avalanche, landslip or other movement of soil), which we used for the linkage of SNC and SLF data bases in the supplement (see table S1). 

9) 142- “two regions defined by distance from the Alps…” Using the <=>25 km distinction adds little to the demographic data as you found a median distance of 61 km from residence to place of death and described victims as predominantly from an Alpine region. This would suggest that all of Switzerland is an Alpine region, which may be true. The 25 km measure offers little useful detail beyond just place of residence and location of death.

Authors` response: Another advantage of the linkage of the SNC and the SLF databases is that we now have information of the place of residence and the exact place of the avalanche for all avalanche victims. On page 7, line 146-149 we introduced the geographical areas, which we used to zone Switzerland. We used a commonly used division of the Alps into four Alpine regions and divided the area North of the Alps (which is not part of the Alps) into 2 areas according to the distance from the North border of the Alps. The majority of the Swiss population (around three quarters, see table 1 in the manuscript) is living in these two areas because most of the settlements are located there. Compared to that around half of the avalanche victims lived in the two non-Alpine regions and the other half in the Alpine regions. The 25 km distance from the Northern border of the Alps was an arbitrary choice to divide the non-Alpine region into 2 areas. 25 km seemed to be a proximity for reaching the Alps within a reasonable time for frequent leisure activities in the Alps. We intended to compare Alpine areas, areas close to the Alps and areas further away from the Alps concerning where the avalanche victims came from. 

10) 159- Table 1. Two time periods 1/1/1995-2000 and 5/12/1995-2014 are given but it is not clear why or how these two periods were chosen. See above for line 96, there is a lack of clarity on how cohorts and time periods were determined. You also list “time periods” in line 160 then report in line 172-3 “two points in time”, both for same table. This is confusing.

Authors` response: We agree that this is confusing. We clarified this in the text and in the table description and column description (see page 9).

---

## [Editor Report · Decision Letter 1]

12 Nov 2019

Snow avalanche deaths in Switzerland from 1995 to 2014 – results of a nation-wide linkage study

PONE-D-19-17260R1

Dear Dr. Berlin,

We are pleased to inform you that your manuscript has been judged scientifically suitable for publication and will be formally accepted for publication once it complies with all outstanding technical requirements.

With kind regards,

Bruno Masquelier, PhD

Academic Editor

PLOS ONE

Additional Editor Comments (optional): All questions and comments from the reviewers have been addressed. Thank you.
---

## [Editor Report · Acceptance letter]

21 Nov 2019

PONE-D-19-17260R1 

Snow avalanche deaths in Switzerland from 1995 to 2014 – results of a nation-wide linkage study 

Dear Dr. Berlin:

I am pleased to inform you that your manuscript has been deemed suitable for publication in PLOS ONE. Congratulations! Your manuscript is now with our production department. 

With kind regards,

on behalf of

Dr. Bruno Masquelier 

Academic Editor

PLOS ONE